# Rainfall analysis in mountain streams affected by torrential floods on Madeira Island, Portugal

Sérgio Lopes[1], Marcelo Fragoso[1], Eusébio Reis[1]

[1] University of Lisbon, Institute of Geography and Spatial Planning, Centre of Geographical Studies, TERRA Associate Laboratory, Edifício IGOT, Rua Branca Edmée Marques, 1600-276, Lisbon, Portugal.
*Correspondence to*: Sérgio Lopes (lopes.sergiodasilva@gmail.com)

**Abstract**

Torrential floods are powerful and destructive events that result from a mix of debris and water moving rapidly down steep channels in mountainous areas. They are normally triggered by heavy rainfall. Predicting these flows is crucial, and analyzing critical rainfall is essential for civil protection early warning systems.

Empirical rainfall thresholds for Madeira Island were established using historical data on torrential floods and rainfall from 2009 to 2021. The analysis differentiates between antecedent rainfall, calculated with a power law for drainage over time, and event-specific rainfall, determined as the maximum 24-hour precipitation during intense rainfall events.

Our findings show that antecedent rainfall plays a significant role in triggering torrential floods on Madeira Island. Calibrated antecedent precipitation over 15 days (110 mm) and 30 days (130 mm), combined with a maximum rainfall of 250 mm in 24 hours, exemplify critical conditions. A strong correlation between maximum precipitation over 24 and 12 hours was also found and can be described by a linear regression model, aiding in predicting critical 12-hour rainfall maximums. Rainfall-runoff modeling also revealed preliminary findings on the relationship between catchment area, sub-daily precipitation peaks, and peak discharge.

As far as we know, this is the first study to introduce combined rainfall thresholds for the occurrence of torrential floods in Madeira Island catchments.

## 1 Introduction

Torrential flows, such as debris flows and debris floods, constitute mixed masses of debris and water that move at high velocities within steep channels in mountainous regions. In areas where these channels are formed by deep and narrow-bottomed valleys, there is often a higher occurrence of interactions between slope movements (such as landslides and debris flows) and fluvial dynamics (such as flash floods). Often, these flows are amplified by shallow translational slides that occur in the upper sectors of slopes and headwaters of catchments (SRES, 2010; Fragoso et al., 2012; Lopes et al., 2020). As these materials descend toward valley bottoms and streamlines, the initially stationary soils are rapidly transformed into a mixture of fine particles (sand, silt, clay) and coarse materials (cobble, boulder). This mixture also includes woody debris and water, resulting in a high-density mass where the solid load often exceeds 50% of the total mass. These materials are displaced by gravitational force, typically through successive impulses (Costa, 1988; Scott, 1988; Zêzere et al., 2005; Fragoso et al., 2012). Ultimately, this material is deposited only in areas with low ground gradient (MLIT, 2004), where flooding can occur. Monitoring-based analysis is crucial for improving our understanding of the mechanisms that trigger torrential flows and their propagation. Rainfall is the most common triggering factor for these events (e.g., Zêzere et al., 2005). Understanding the specific rainfall conditions that lead to torrential flows is critical for providing timely early warnings related to such phenomena. The antecedent rainfall and the soil moisture conditions, may influence the triggering of torrential flows (Oorthuis et al., 2021; Oorthuis et al., 2023). In this sense, when analysing critical rainfall conditions for the initiation of torrential flows, it is important to differentiate between event rainfall and antecedent rainfall (Schröter et al., 2015). However, previous research have primarily focused on establishing specific rainfall thresholds for the initiation of torrential flows, emphasizing the impact of rainfall duration and intensity (Abancó et al., 2016).

Indeed, the temporal variation of rainfall, including antecedent rainfall (over days and weeks), as well as the duration and intensity of heavy rainfall, significantly influences the occurrence of torrential flows and other related disasters. The impact of antecedent rainfall on terrain is a complex process which can lead to soil saturation. Normally, the rain that is retained on the ground in a given area on any given day decreases over time due to the drainage process. This effect can be quantified applying a power law, which accounts for the draining of early precipitation and accumulation of late rainfall (Crozier, 1986; Glade et al., 2000). However, few studies have focused on analysing the combinations between antecedent rainfall and maximum values of heavy rain events for empirical rainfall thresholds related to torrential floods. It is true that several studies have focused on the study of intense rainfall on Madeira Island, especially after the great flood of 20 February 2010 (Luna et al., 2011; Fragoso et al., 2012; Gorricha et al., 2012; Couto et al., 2012; Levizzani et al., 2013). However, there is still limited concrete information regarding critical rainfall thresholds.

Madeira, located in the North Atlantic, is a volcanic island 600 km northwest of the African coast. It is a relatively small mountainous territory with an elongated shape, covering 741 km². Known for its stunning landscapes, Madeira boasts V-shaped valleys, steep slopes, and rugged sea cliffs. The island's terrain is predominantly above 500 m (90% of its surface), with 35% above 1000 m, reaching a maximum altitude of 1862 m at Pico Ruivo.

The risk of torrential floods in Madeira is very high due to the high density of streamlines and the large population (250,744 inhabitants), with 42% of the total population concentrated in the municipality of Funchal and 93% residing on the south side of the island. Recent economic growth, new road access, and urban sprawl can contribute to the occupation of hazard areas, also on the northern side of the island. As a result, the risk of torrential floods in these rural areas may increase.

The primary objective of this study is to determine critical precipitation threshold values that can lead to torrential floods. To achieve this, the analysis is based on historical torrential events and past rainfall measurements. Specifically, data from rainfall events that triggered torrential events will be compared with data from those that did not. A secondary objective is to discuss aspects related to the uncertainty surrounding the definition of specific rainfall conditions and mechanisms that trigger torrential floods in the catchments of Madeira Island.

Establishing accurate rainfall thresholds for torrential floods early warning systems is essential, but the lack of standardized procedures poses a challenge. The proposed criteria consider antecedent rainfall conditions along with maximum daily and sub-daily rainfall amounts.

## 2 Data and methods

The methodology employed in this research is depicted in Fig. 1. The subsequent subsections detail the specific methods and data utilized. The precipitation data sets used in this study are presented in Table 1. Figure 2 is essential as it illustrates the study catchments, the inventory of heavy rainfall-torrential floods and the rain-gauge stations utilized in this research.

### 2.1 Inventory methodology

Madeira island contains approximately 126 catchments, 94% of which are less than 25 km² in area. The island experiences a diverse spatial distribution of heavy rainfall events. Some of these events can trigger torrential floods, while others may lead to increased streamflow discharges across the island without causing overbank flows.

The inventory of precipitation-torrential flood events (Tables 3 and 4) was compiled using one of the following selection criteria based on the occurrence of: a) interactions between slope instability and stream flow; b) people affected (evacuated, displaced, or injured); c) at least one victim (deceased or missing); and d) damage to at least one public or private infrastructure. The photographic record illustrates the destructive effects that occurred in local settlements affected by torrential floods (Fig. 3).

Critical thresholds were determined using a sample of rainfall data from selected rain gauges located on the mountain (south side of the island above 500 meters) and on the north side (regardless of altitude), as shown in Table 1. This rule of spatial identification facilitated the selection of rain gauges in the catchment sectors where torrential flows typically originate. The goal is to identify a combination of critical thresholds which, when surpassed at one or more rain gauges, can trigger the torrential flood early warning systems.

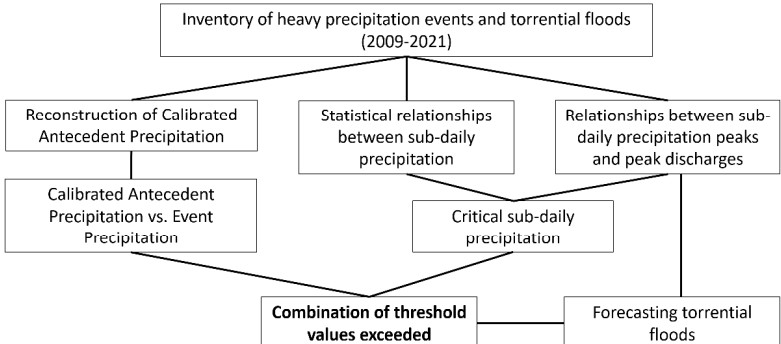

Fig. 1 Methodology for determining probable rainfall thresholds related to torrential floods and consequent peak discharges.

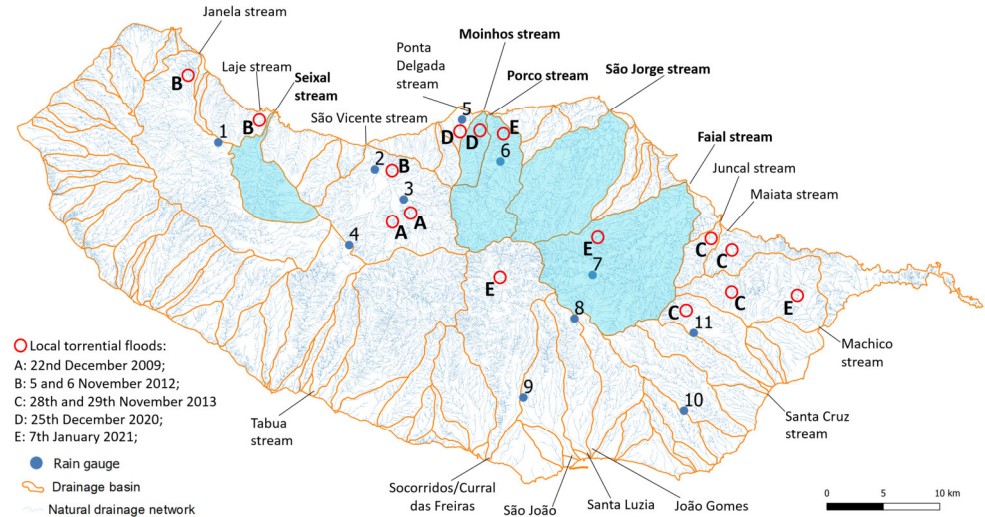

Fig. 2 Madeira Island hydrographic system. The catchments for the hydrological study are highlighted in blue, while the red circles indicate localities affected by torrential floods from 2009 to 2021.

Rain-gauges used in this research: 1 Fanal; 2 Posto Florestal São Vicente; 3 Achada do Til; 4 Bica da Cana; 5 Ponta Delgada; 6 Fajã do Penedo; 7 Fajã da Nogueira; 8 Areeiro; 9 Trapiche; 10 Camacha; 11 Santo da Serra.

Table 1 Rainfall data-sets used in this work (see location in Fig. 2).

| Heavy rainfall events | Purpose of data (*) | Rain-gauges | Geographic position in the Island and altitude |
|---|---|---|---|
| 22 Dec. 2009 (torrential floods) | A | Achada do Til | North side (349 m) |
| 2 Feb. 2010 (torrential floods) | A | Areeiro<br>Camacha | Mountain top (1583 m)<br>South side (644 m) |
| 20 Fev. 2010 (torrential floods) | A | Trapiche<br>Areeiro<br>Camacha | South side (536 m)<br>Same as above<br>Same as above |
| 21 Oct. 2010 | B | Trapiche/ Areeiro/ Camacha | Same as above |
| 25 Nov. 2010 | B | Trapiche/ Areeiro/ Camacha | Same as above |
| 20 Dec. 2010 | B | Trapiche/ Areeiro | Same as above |
| 25 Jan. 2011 (torrential floods) | A | Trapiche/ Areeiro/ Camacha | Same as above |
| 5 e 6 Nov. 2012 (torrential floods) | A | Posto Florestal de São Vicente | North side (120 m) |
| 28 e 29 Nov. 2013 (torrential floods) | A | Santo da Serra | South side (880 m) |
| 25 Dec. 2020 (torrential floods) | A/ C | Fajã do Penedo | North side (259 m) |
| 7 Jan. 2021 (torrential floods) | A/ C | Fajã Penedo<br>Fajã da Nogueira | Same as above<br>North side (621 m) |
| | | | |
| **Years without heavy rainfall events** | | **Rain-gauges** | **Geographic position in the Island** |
| 2011/2012; 2014/2015; 2015/2016; 2016/2017; 2018/2019; 2019/2020; 2021/2022 | A | Posto Florestal de São Vicente<br><br>Areeiro | Same as above<br><br>Same as above |

*Purpose of data: A – Calibrated Antecedent Precipitation (CAP) and statistical relationships between sub-daily precipitation; B – Statistical relationships between sub-daily precipitation; C – Relationships between sub-daily precipitation peaks and peak discharges.

### 2.2 Case Studies of Specific Events (Dec 2020 - Jan 2021)

Most of the events analyzed here have resulted in small local torrential floods, primarily occurring on the northern side of the island. Consequently, the analysis of the relationships between sub-daily precipitation peaks and peak discharges, as well as the hydrological rainfall-runoff case study and the estimation of peak discharges, were conducted in a group of catchments on the north side of the island (Fig. 1). Particular focus was given to two specific events on December 25, 2020, and January 7, 2021, due to their higher temporal resolution records (10 minutes) at various measurement points. These heavy rain events, occurring within a 15-day timeframe, provided valuable data, which was collected from the Fajã da Nogueira, Fajã do Penedo, São Vicente, and Fanal rain gauges (Fig. 2). For the heavy rainfall events and torrential floods that occurred prior to these dates, as listed in Table 3, there is insufficient reliable data to conduct the analysis properly.

### 2.3 Definition of heavy precipitation event

According to the methodological flow shown in Figure 1, it's crucial to clarify the concept of event precipitation. Precipitation occurring during an event and its duration can be determined using various criteria. For example, the Portuguese Meteorological Institute (IPMA) uses a rule of thumb that considers total precipitation exceeding 30 mm in a 6-hour moving interval, resulting in an average of 18 days of heavy precipitation annually in mountainous areas like Areeiro (see location in Figure 2). Another criterion specifies that an event begins when hourly precipitation reaches or exceeds 4 mm and ends when it falls below this threshold for a consecutive 6-hour period. Typically, most heavy rainfall events are concentrated within a 24-hour timeframe (Lopes, 2015).

In this analysis, event precipitation corresponds to the maximum 24-hour precipitation, calculated as a sliding sum over the days of heavy rainfall. When analyzing the influence of precipitation in the days preceding a specific rainfall event, it is logical to separate event rainfall from antecedent precipitation (Schröter et al., 2015). This criterion is also applied here. All calculations are based on 10-minute readings (in millimeters), available for most events through the regional network of rain gauge stations.

### 2.4 Calibrated antecedent precipitation method

Precipitation from previous days can significantly or partially influence the triggering of slope movements and torrential floods. A paradigmatic example of this influence occurred during the intense rainfall event on February 20, 2010, which caused torrential floods in Madeira during a particularly anomalous wet winter (Fragoso et al., 2012). However, the overland flow following the rain, diminishes over time due to surface drainage (Zêzere et al., 2005). To assess the impact of rainfall in the days and weeks preceding slope instability, Crozier (1986) proposes introducing an exponential function to express the diminishing significance of rainfall with increasing temporal distance from the date of interest.

The mathematical expression used to calculate calibrated antecedent precipitation (Zêzere et al., 2005, in Crozier, 1986) is as follows:

$$CAPx = K\,P1 + K^2\,P_2 + ...K^n\,P_n$$

where CAPx is the calibrated antecedent precipitation for day x; P1 is the daily rainfall for the day before day x; Pn is the daily rainfall for the n-th day before day x. K is an empirical parameter typically considered between 0.8 and 0.9, depending on the draining capacity of the material and the hydrological characteristics of the area. In this study, we adopt the value K = 0.9, based on references from existing literature (Zêzere et al., 2005; Marques et al., 2008). This equation makes negligible precipitation occurred more than 30 days before a torrential flood event. Consequently, the reconstitution of the calibrated antecedent precipitation was only calculated for durations of 5, 10, 15, and 30 days.

In theory, with the empirical parameter K = 0.9, we assume that 90% of the rainwater that fell on a certain day prior to the day of interest was temporarily accumulated in the soils of the catchment-stream system. This evaluation considers rainwater from previous days and weeks, which can contribute to soil saturation and slope instability. The remaining 10% of the water was quickly drained into the water network, percolated, or evaporated. It is important to note that K serves as a sensitivity parameter, and choosing a different value of K would significantly influence and alter the proposed thresholds.

## 2.5 Relation between Calibrated Antecedent Precipitation and Maximum 24-hour Precipitation

As shown in Figure 1, the characteristic combinations of precipitation from preceding days and heavy rain events were investigated by examining the relationship between the maximum precipitation in 24 hours (Pmax24h) and the corresponding antecedent precipitation calibrated for different periods (5, 10, 15, and 30 days).

For this analysis, a set of eight episodes of heavy rainfall-torrential floods, from 2009 to 2021 was considered (Table 1). A data sample was collected from rain gauges located near the areas of the island where rainfall was significantly higher during those episodes. One of the conditions followed in setting up this sample was to ensure that there were no data gaps in the previous weeks, allowing for a more rigorous calculation of the calibrated antecedent precipitation. For some heavy rainfall events, rainfall data from more than one rain-gauge was used.

In the compilation of these climate series, rainfall data from a set of years without records of heavy rainfall events were also included (Table 1 and Fig. 4). The identification of critical thresholds requires a statistical analysis of precipitation data from years with and without torrential flood records, to detect the existence of a clear separation in precipitation behaviour between both samples (Zêzere et al., 2005).

The precipitation thresholds were determined using a mosaic of site-specific local rainfall data. A similar procedure was adopted by Segoni et al. (2014). Given the general characteristics of the Madeira territory, this approach appears to be more effective than using a single regional threshold derived from just one rain gauge.

In years without records of torrential floods, the precipitation values needed to analyse the relationship between daily precipitation and antecedent precipitation were obtained using the maximum annual precipitation technique for various durations. When constructing these extreme value series, multiple values per year can be utilized. This becomes particularly relevant when two heavy rainfall events followed by floods occur within the same hydrological year.

**2.6 Data for different sub-daily durations**

In this section, a comparative analysis was conducted on the relationship between the maximum precipitation over 24 hours (Pmax24h) and the maximum precipitation over shorter durations (Pmax12h, Pmax6h, Pmax3h, and Pmax1h), following the methodological flow presented in Figure 1. This analysis aimed to evaluate the statistical significance of the dependencies between these variables and predict whether there will be a higher or lower sub-daily concentration of rain during heavy rainstorm forecasts.

To conduct this study, we formed a dataset from nine rain gauges located on the north side of the island, on the mountain top and at the south side above 500 m altitude (rain gauges 1, 2, 3, 6, 7, 8, 9, 10, and 11, as shown in Fig. 2). This dataset covered a total of eleven heavy rainfall events that occurred between 2009 and 2021, including those that caused torrential floods and others that led to a sudden and significant increase in streamflow discharges without overbank flows (Table 1). By incorporating data from different rain gauges in the same sample, we aimed to emphasize the importance of identifying characteristic values that represent the climatology of intense rainfalls in those concrete sectors: the entire north side of the island, the top of the mountain and the south side above 500 altitude. The occurrence of heavy rainfall and the previously accumulated rainfall in these areas play a crucial role in understanding the occurrence of floods.

**2.7 The hydrological model**

The estimation of peak discharges was conducted for five catchments situated on the northern side of Madeira Island: Faial, São Jorge, Porco, Seixal, and Moinhos streams (Fig. 2 and Table 2). Rainfall data from the January 7, 2021 event, collected from three rain gauges (Fajã da Nogueira, Fajã do Penedo, and Fanal, numbered 7, 6, and 1 in Fig. 2, respectively), with 10-minute recording intervals, were used as input parameters for the hydrological model.

The respective hyetographs were prepared using the hourly right time precipitation values. The flood hydrographs and peak flows were calculated under the assumption that the amount of rainfall recorded at each rain-gauge station uniformly affected the catchments where they are located. The estimated values pertain to the outflow at the mouth of the catchments.

Flood hydrographs are determined based on the hyetograph data mentioned earlier and the synthetic unit hydrograph from the Soil Conservation Service (S. C. S.). This calculation is performed using the HEC-HMS model (Hydrologic Engineering Center - Hydrologic Modeling System) developed by the U.S. Army Corps of Engineers (version HEC-HMS 4.8).

The S. C. S. unit hydrograph is characterized by the catchment's response time to peak precipitation, as given by the expression: $tP = 0{,}6 \times tC$, where $tP$ is the lag time and $tC$ is the time of concentration of the catchment. The concentration and lag times for the studied catchments are shown in Table 1. To estimate the volume of precipitation water transformed into flow, the lag time method was employed.

The curve number (CN), an empirical parameter developed by the Soil Conservation Service (SCS), was used to classify permeability in the catchments studied (McCuen, 1982; Leal, 2012; Lopes, 2020). The low degree of permeability in Madeira's catchments is primarily due to natural conditions, which are largely influenced by the characteristics of volcanic

rocks and soils (Leal et al., 2020). Considering the prevalence of agricultural and forestry areas, and referring to the S. C. S.
tables, it was decided to adopt a common CN of 81 for all catchments in this hydrological modelling exercise. This choice
corresponds to AMCIII conditions (completely saturated soil).

Table 2: Physical parameters of the catchments under study.

|  | *Faial* | *São Jorge* | *Porco* | *Seixal* | *Moinhos* |
|---|---|---|---|---|---|
| Catchment area (km²) | 50 | 32 | 20.2 | 14.1 | 5.2 |
| Length of main stream (km) | 14.3 | 10.4 | 10.2 | 10.3 | 5.5 |
| Average gradient of main stream (km/km) | 0.154 | 0.145 | 0.154 | 0.153 | 0.249 |
| Time of concentration (hours) | 3h27 | 2h34 | 2h30 | 2h31 | 1h26 |
| Lag time (hours) | 2h04 | 1h32 | 1h30 | 1h31 | 0h51 |

Table 3 Rainfall-to-runoff torrential floods and their calibrated antecedent precipitation (CAP).

| Date | Rain-gauges | Maximum precipitation in 24 hours | Calibrated antecedent precipitation | | | | |
|---|---|---|---|---|---|---|---|
|  |  |  | **3 days** | **5 days** | **10 days** | **15 days** | **30 days** |
| 22 December 2009 | Achada do Til | 290.8 | 131.9 | 259.7 | 348.6 | 374.5 | 385.6 |
| 20 Fev. 2010 | Areeiro | 333.8 | 142.2 | 155.6 | 213.5 | 216.3 | 291.6 |
| 5 and 6 November 2012 | Posto Florestal São Vicente | 317 | 57.2 | 57.3 | 90.5 | 117.8 | 134 |
| 28 and 29 November 2013 | Santo da Serra | 241 | 20.3 | 27.3 | 38.9 | 40.6 | 53.1 |
| 25 December 2020 | Fajã do Penedo | 335.5 | 43.7 | 43.7 | 49.9 | 53.7 | 83.7 |
| 7 January 2021 | Fajã do Penedo | 475 | 33.9 | 33.9 | 41.9 | 158.1 | 162.1 |

Table 4 Maximum rainfall for different sub-daily intervals, in moving time window.

| Flood events | Rain-gauges | Pmax24h | Pmax12h | Pmax6h | Pmax3h | Pmax1h |
|---|---|---|---|---|---|---|
| 22 December 2009 | Achada do Til | 290.8 | 197 | 179.6 | **150.4** | **107.8** |
| 20 Fev. 2010 | Areeiro | 333.8 | 315.2 | 224.2 | 139.4 | 64.6 |
| 5 and 6 November 2012 | Achada do Til | 211 | 146.6 | 93 | 54 | 26.2 |
| | Posto Florestal São Vicente | 317 | 229.8 | 121.4 | 78.4 | 32.8 |
| 28 and 29 November 2013 | Santo da Serra | 241 | 150.8 | 126.6 | 101.4 | 43.2 |
| 25 December 2020 | Fajã do Penedo | 335.5 | 213.5 | 137 | 106.5 | 47 |
| 7 January 2021 | Fajã do Penedo | **475** | 362 | **212.5** | 116 | 43 |
| | Fajã da Nogueira | 287.4 | 177 | 115.8 | 81.6 | 39.4 |
| | Fanal | **471.6** | **379.4** | 195.4 | 117.8 | 45.4 |
| **Average** | | **329.2** | **241.3** | **156.2** | **105.1** | **49.9** |

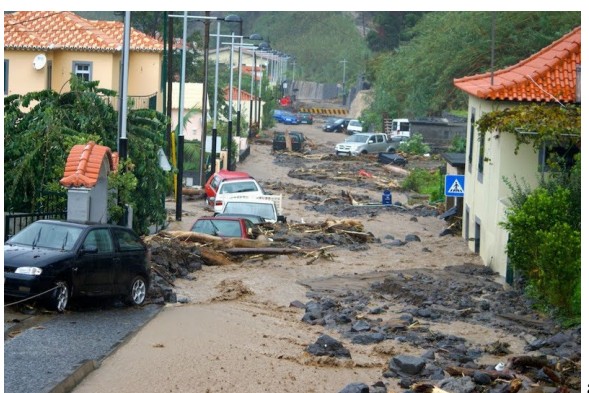 a)
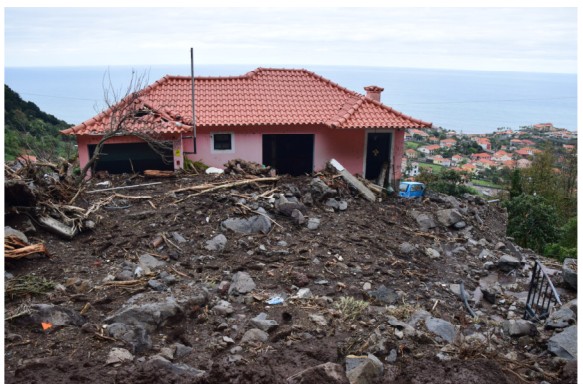 b)

Fig. 3 Destructive effects caused by torrential floods on the north
side of Madeira Island: a) 5 and 6 November 2012; b) 25 December 2020.

## 3 Rainfall thresholds

As illustrated in Figure 1, the methodological approach involved compiling an inventory of rainfall-runoff events, with particular attention to maintaining the reliability of the rainfall time series databases. This study used data from a series of eleven heavy rainfall events that occurred between 2009 and 2021. However, not all of these events gave rise to torrential

floods, as shown in Table 1. Five of the six events inventoried in Table 3 (except for 20 February 2010), along with their respective occurrences mapped in Fig. 2, illustrate that during certain rainfall storms, subsequent floods tend to have a localized spatial impact.

These events primarily occurred on the northern side of the island, characterized by rocky and mountainous terrain. Vertical erosion has shaped ravines and exceptionally deep valleys within the island's interior. Field surveys conducted immediately

after the torrential flood disasters allowed for the georeferencing of 10 streams affected between 2009 and 2021. These specific five events were chosen for analysis because they caused significant damage in various locations. These areas experienced the displacement of people and material damage, impacting both private property and public infrastructure. The other heavy rain events recorded caused minor damage.

### 3.1 Calibrated antecedent precipitation

The dots shown in Figure 4 represent values from a set of known heavy rainfall events and a sample of the yearly maximum 24-hour rainfall from years without heavy rainfall events. This step was adopted to analyze the relationship between the calibrated antecedent rainfall and the rainfall of the event, as illustrated in Figure 1. For the sample of years with no record of intense precipitation events-torrential floods, the maximum values of each hydrological year were considered. This includes

the maximum 24-hour precipitation and the corresponding calibrated antecedent precipitation (CAP) value (blue dots), as well as the maximum CAP and the corresponding 24-hour precipitation value (black dots).

For some heavy rain events, data from more than one rain gauge was included in this graphical analysis. In all graphs it is evident that maximum rainfall pairs for years without recorded torrential floods (represented by the blue and black dots) consistently fall below the horizontal line corresponding to the maximum 24-hour rainfall of 250 mm. Conversely, most of

the points in pairs related to known heavy rainfall events are located close to or above the line (indicated by orange points). This observation confirms that the value of maximum rainfall in 24 hours can be used as a probable threshold for the occurrence of torrential floods on Madeira Island. This threshold can be combined with the corresponding values of antecedent rainfall calibrated for different durations in days, as shown below.

The graphical information relating the maximum precipitation in 24 hours (Pmax24h) to the antecedent precipitation

calibrated for different durations (5, 10, 15, and 30 days) indicates a pattern of data grouping (Fig. 4). The positioning of the vertical line on each graph seeks to identify, from the sample data pattern, the calibrated antecedent precipitation threshold,

above which, torrential flood events may occur. Additionally, Table 3 presents the calibrated antecedent precipitation values for different durations from a set of six case studies.

In order to define the number of days that are most relevant to antecedent precipitation, the best results correspond
to the CAP at 15 and 30 days, where the graphs show only two orange dots positioned to the left of the respective vertical lines of calibrated antecedent precipitation (Figs. 4c and 4d). However, even for the other durations (5 and 10 days), there is a tendency for points to be more concentrated to the right of the respective vertical lines (Figs. 4a and 4b). Therefore, all the combinations obtained can be used as probable thresholds for torrential floods on Madeira Island. Table 5 presents a proposed combination of rainfall thresholds associated with different probabilities of torrential floods occurring.

The information summarized in Fig. 5 pertains to a specific case involving two events of heavy rainfall-to-runoff torrential floods. These events occurred within the same hydrological year (2020/2021). In this particular scenario, the calibrated antecedent precipitation (CAP) at 5 and 15 days on the date of the first event (December 25, 2020) was relatively low. During the second event (January 7, 2021), the precipitation from the first event significantly influenced the high value of the CAP at 15 days (158 mm). However, the January 7, 2021, torrential event has even higher Pmax24 than the previous
event (Table 3). Thus, in this specific case, it is not possible to ascertain the exact contribution of CAP15 (*Fajã do Penedo* rain-gauge) in triggering the local flood on January 7, 2021. This case involves two heavy rainfall events occurring within a short span of less than 15 days. Nonetheless, it demonstrates the significance of accumulated rainfall over the preceding days and weeks in saturating the soil with water, and highlights the consequences this can have in terms of land instability and the occurrence of torrential floods.

Table 5 Proposed criteria for classifying probability of occurrence of torrential floods
in Madeira Island's catchments.

| Relationships between rainfall for different durations | Probable thresholds of maximum rainfall pairs (*) | Probability of torrential floods |
|---|---|---|
| CAP5 days – Pmax24 h + Critical Pmax 12 h **(a)** | - CAP 5 days > 80 mm; - Pmax24 h > 250 mm and Pmax12 h > 205 mm; | **1) Pre-critical soil saturation condition** (exceedance of CAP thresholds for different durations); |
| CAP10 days – Pmax24 h + Critical Pmax 12 h **(b)** | - CAP 10 days > 90 mm; - Pmax24 h > 250 mm and Pmax12 h > 205 mm; | |
| CAP15 days – Pmax24 h + Critical Pmax 12 h **(c)** | - CAP 15 days > 110 mm; - Pmax24 h > 250 mm and Pmax12 h > 205 mm; | **2) High probability of flooding based on event precipitation** (Pmax(24h) > 250 mm and Pmax(12h) > 205 mm); |
| CAP30 days – Pmax24 h + Critical Pmax 12 h **(d)** | - CAP 30 days > 130 mm; - Pmax24 h > 250 mm and Pmax12 h > 205 mm; | **3) Very high probability of flooding based on the combination of CAP and event precipitation** (a) or b) or c) or d) |

**(*)** Condition that critical thresholds occur in at least one rain gauge of Table 1, located either in the mountains (including the south side of the island above 500 meters above sea level) or on the north coast of Madeira Island (regardless of altitude).

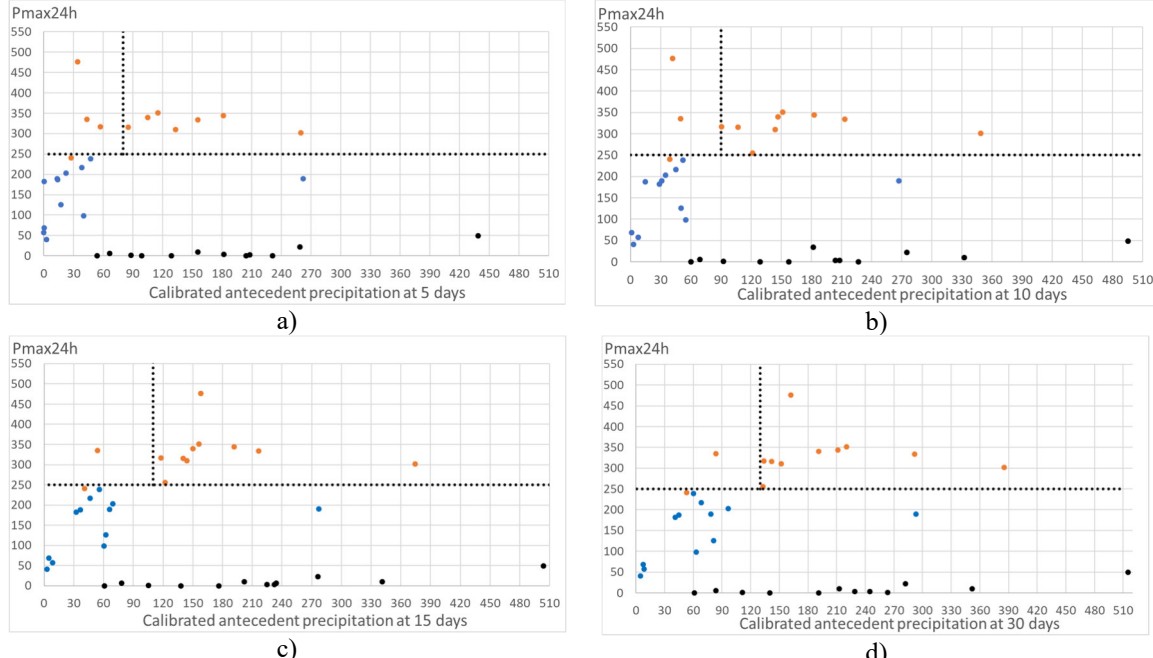

Fig. 4 Relation between maximum 24-hour precipitation and calibrated antecedent precipitation at 5 days (a), 10 days (b), 15 days (c) and 30 days (d). The orange dots refer to the values associated with heavy rainfall events between 2009 and 2021. The blue dots correspond to values obtained from the yearly maximum 24-hour rainfall, while the black dots represent the yearly maximum calibrated antecedent precipitation, computed for years without torrential floods.

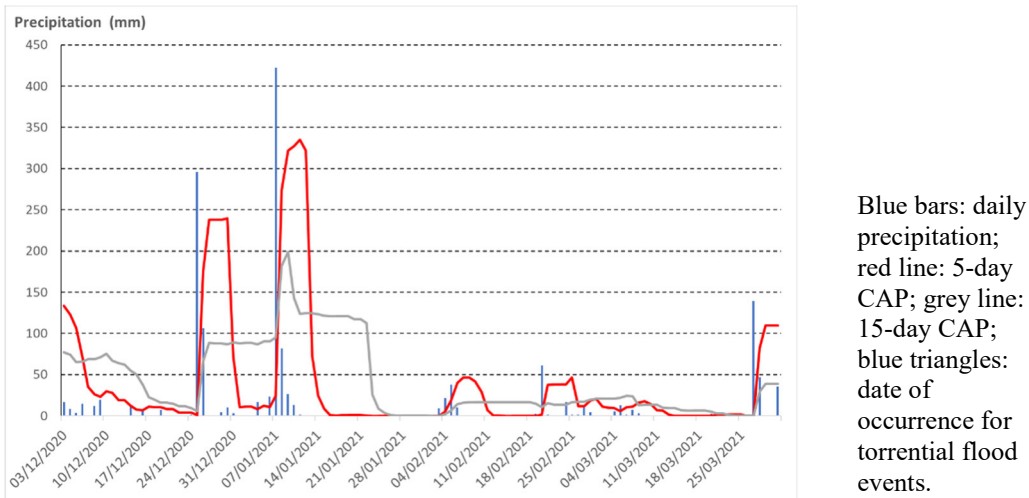

Blue bars: daily precipitation; red line: 5-day CAP; grey line: 15-day CAP; blue triangles: date of occurrence for torrential flood events.

Fig. 5 Daily precipitation and calibrated antecedent precipitation (CAP) at 5 and 15 days, between December 2020 and March 2021, in *Fajã do Penedo*.

## 3.2 Relationships between precipitation of different sub-daily durations

This correlation is crucial for achieving one of the objectives set by the methodology presented in Figure 1, as the concentration of sub-daily precipitation directly influences the flood flow regime in mountain streams. Data from heavy rainfall events that have led to floods indicate that more than 80 percent of the maximum daily rainfall (in 24 hours) is concentrated in less than 12 hours. The results reveal a strong correlation ($R^2 = 0.83$) that can be mathematically described by a linear regression model between Pmax24h (maximum precipitation over 24 hours) and Pmax12h (maximum precipitation over 12 hours), following the equation: *Pmax12h = 0.7553\*Pmax24h + 16.257* (Fig. 6). These indicated relationship allow us to estimate the probable critical maximums within a 12-hour period, based on the predicted precipitable water values over 24 hours. Interestingly, there is no significant correlation between Pmax24h and the maximum precipitation for shorter durations: 6 hours ($R^2 = 0.57$), 3 hours ($R^2 = 0.40$), and 1 hour ($R^2 = 0.12$). However, for mountainous areas on the northern side of Madeira Island, certain average values of heavy rainfall can be inferred (Table 4).

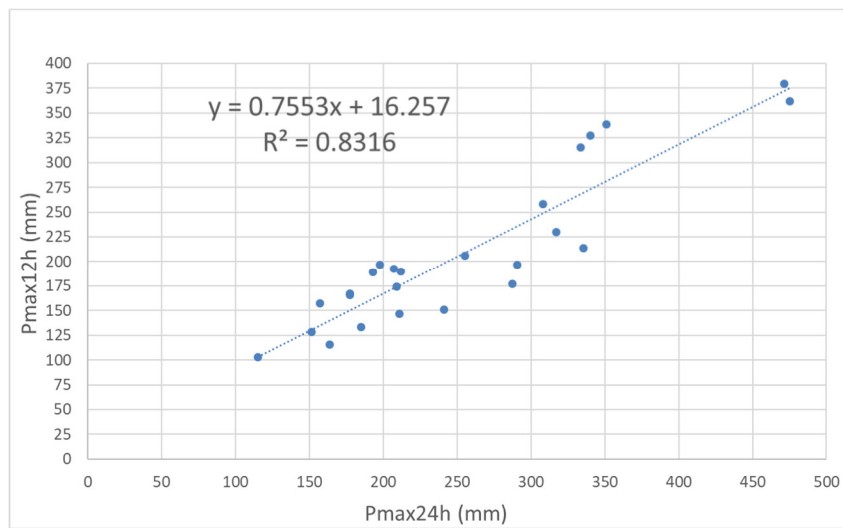

Fig. 6 Relation between precipitation in 12 h (Pmax12h) and 24 h (Pmax24h), using data from events from 2009 to 2021.

## 4 Relationships between sub-daily precipitation peaks and peak discharges

In the previous subsection, we observed that heavy rainfall events leading to torrential floods are associated with more than 80 percent of the maximum 24-hour rainfall occurring in less than 12 hours. In this subsequent step of the methodology depicted in Figure 1, the sequences of critical hours of heavy rain and the resulting peak discharges in catchments of varying

sizes are analyzed. Peak discharges were estimated to measure the hydrological response of catchments to specific heavy rainfall events, specifically on December 25, 2020, and January 7, 2021.

On December 25, 2020, there was particularly intense rainfall on the northern coast of the island, specifically in the streams of Ponta Delgada and Moinhos (Fig. 2). The records from the *Fajã do Penedo* rain gauge clearly indicate the exceptional precipitation during this event, with 24-hour maximum exceeding 300 mm (Table 4). The hyetograph of *Fajã do Penedo* (Fig. 7a) reveals abundant rainfall during the first 15 hours of the day, totalling 136.5 mm. Subsequently, a critical period of heavy rainfall occurs in the afternoon, with 124 mm recorded over 4 hours (Fig. 7a). During this time, several slope movements occurred in the Ponta Delgada catchment (Fig. 2). For the heavy rain event of January 7–8, 2021, it is notable that there were seven non-consecutive hours with hourly precipitation exceeding 30 mm (Fig. 7b).

The studied catchments vary in size, ranging from 5.2 km² (Moinhos) to 50 km² (Faial), and exhibit a generally elongated shape (Fig. 2). Their physical parameters are detailed in Table 2. The lengths of the respective main streams are relatively short, with the longest being 14.3 km (Faial) and the shortest measuring 5.5 km (Moinhos). All catchments share a common feature: the predominance of steep slopes and a short time of concentration, typically about 3 hours or less (as shown in Table 2). The topographic conditions of these catchments, along with their land use characteristics, play a crucial role in the occurrence of torrential floods (Lopes et al., 2020).

The case study selected for estimating peak discharges was the heavy rainfall event of January 7, 2021. During this event, a longer sequence of precipitation records was available at a 10-minute interval, without any data gaps. The HEC-HMS model estimated peak flows for the mouth of the Faial catchment (50 km²) and the São Jorge catchment (32 km²), reaching 297 m³/s and 215.6 m³/s, respectively. In the Porco catchment (20 km²), the peak flow at the mouth was 204 m³/s, while at Seixal (14 km²), it reached 137 m³/s. Notably, the differences in peak flows were 81.4 m³/s (Faial vs. São Jorge) and 148 m³/s (Porco vs. Moinhos). Additionally, the time intervals between flood peaks varied: 35 minutes (São Jorge vs. Faial) and 29 minutes (Moinhos vs. Porco).

The hydrograms of the Faial and São Jorge streams (Fig. 8a) stand out due to the occurrence of several peak discharges, often associated with secondary peaks of intense rainfall. This flow pattern, which varies over time, is indicative of a complex flood event whose duration exceeds that of a simple flood. These secondary peaks of intense rainfall tend to increase the instability of the terrain in the upper sectors of the catchments.

The results obtained contribute to the creation and maintenance of a hydrological database for Madeira Island's streams, specifically focusing on the relationship between daily and sub-daily precipitation maxima and the corresponding peak flows expected at the stream mouth. This relationship is primarily influenced by the area of the respective catchments. For instance, during the January 7, 2021 event, a maximum 24-hour rainfall of 287.4 mm, with an hourly peak of almost 40 mm within the Faial catchment (Fajã da Nogueira rain gauge), resulted in a maximum discharge of 297 m³/s in the terminal stream section (Table 6). It should be noted that all the peak flows shown in Table 6 primarily led to near-flood situations in their respective catchments. In fact, a torrential flood occurred only in the Moinhos catchment (Fig. 2), resulting in significant material damage.

Table 6 Peak precipitation and subsequent peak flows occurred at the catchment outlet
during the heavy rainfall event on January 7, 2021.

| Pmax24h | Pmax12h | Rain-gauge | Catchment | Catchment area (km²) | Peak flow* (m³/s) |
|---|---|---|---|---|---|
| 475 | 362 | Fajã do Penedo | São Jorge | 32 | 216 |
| | | Fajã do Penedo | Porco | 20.2 | 204 |
| | | Fajã do Penedo | Moinhos | 5.2 | 56 |
| 471.6 | 379.4 | Fanal | Seixal | 14.1 | 137 |
| 287.4 | 177 | Fajã Nogueira | Faial | 50 | 297 |

\* Peak flow at the catchment outlet. HEC-HMS estimations.

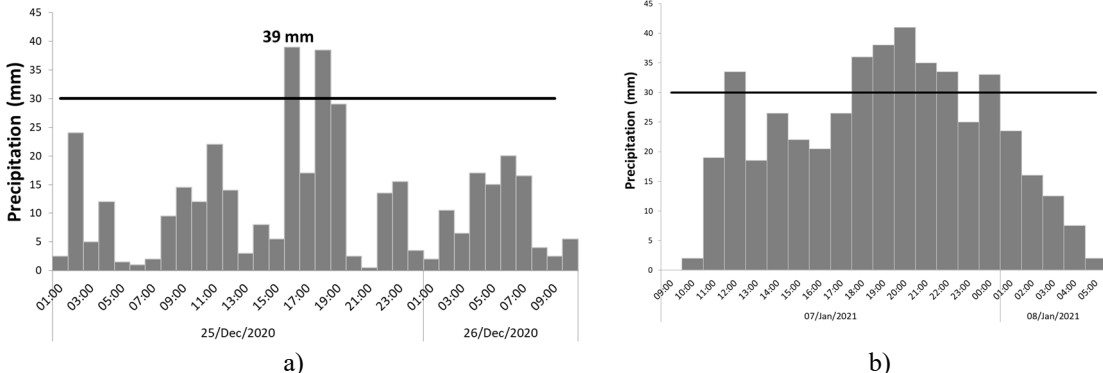

a)                                                            b)

Fig. 7 Hyetograph of *Fajã do Penedo* during the heavy rainfall events of:
(a) December 25, 2020; (b) January 7 and 8, 2021.

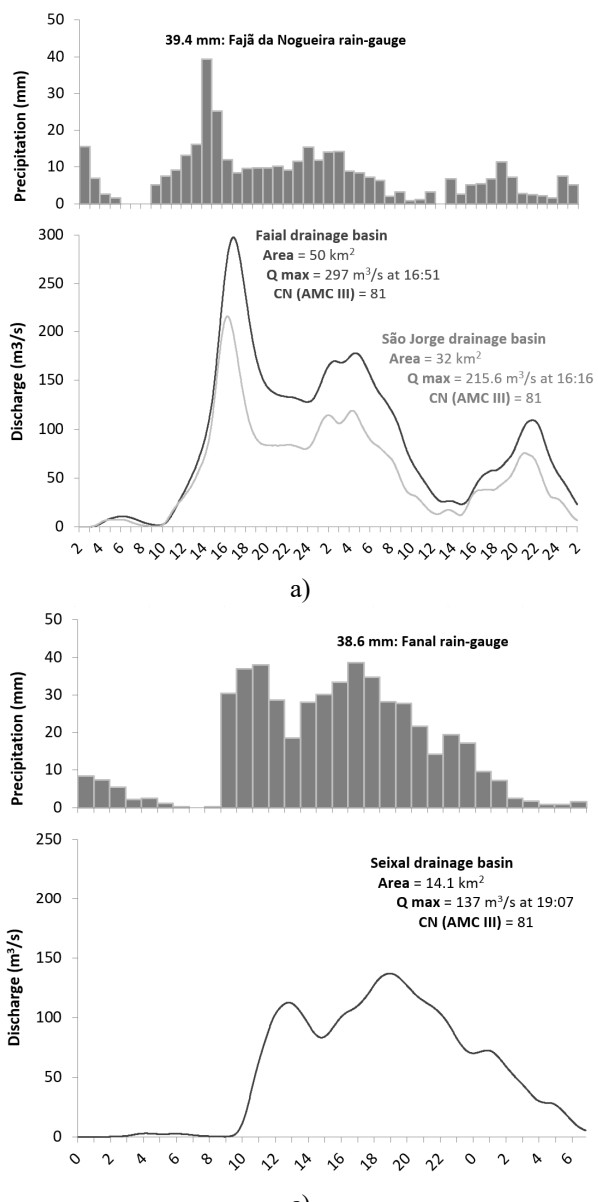

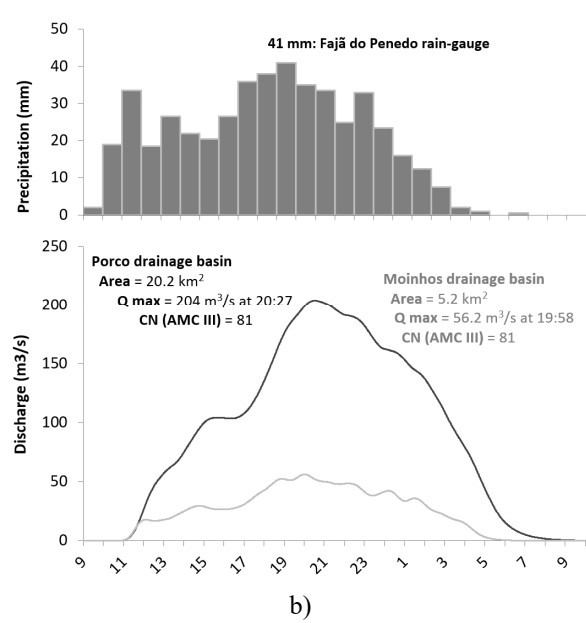

Fig. 8 Hyetographs of *Fajã da Nogueira*, *Fajã do Penedo* and *Fanal* rain-gauges, during the heavy rainfall event on January 7 and 8, 2021 and flow hydrographs of the catchment in the areas of influence of these rain-gauges.

## 5 Discussion

The study focuses on specific areas of the island that are most critical for understanding torrential floods, particularly higher altitudes and the north side. Critical thresholds are determined using a sample of rainfall data from selected rain gauges located on the mountain, specifically on the south side of the island above 500 meters, and on the north side (regardless of altitude). This approach ensures that the critical thresholds obtained for different durations reflect the behavior of heavy rainfall in these specific areas, corresponding to the sectors of the catchments where torrential flows generally begin. This deliberate method allows for the determination of a combined regional threshold, rather than relying solely on a single threshold from just one rain gauge.

The amount of rainfall in the days and weeks preceding a heavy rainfall event can influence the occurrence of torrential floods. Similarly, the concentration of precipitation during the event may determine whether floods occur. Our contribution to the research community was to integrate various precipitation thresholds into a combined set of threshold values. If these values are exceeded (Table 5), they are associated with high or very high probabilities of flooding.

In Madeira Island, the characteristics of volcanic rocks, soils and vegetation favor soil surface water retention over days and weeks, leading to increased slope instability. This hydrological behaviour makes preceding rainfall conditions a crucial variable in determining whether torrential flood events are triggered or not.

The calibrated antecedent precipitation was applied in an attempt to evaluate the critical combinations between precipitation from preceding days and the event-specific precipitation on Madeira Island. Based on the available data, it was possible to determine, with some degree of confidence, a strong relationship between rainfall patterns and the occurrence of local torrential floods. In some cases, the precipitation of a given event, combined with the calibrated antecedent precipitation from the preceding 15 days, can easily exceed the critical threshold of 400 mm. Consequently, some torrential flood events on Madeira Island appear to be linked to the combination of precipitation during a preparatory period of soil instability (up to 15 or even 30 days before) and the intense, short-duration rainfall during the event (typically concentrated in less than 24 hours). However, in some cases (like the torrential flood of 25 december 2020), this influence is less significant, and the event-specific rainfall can be sufficiently high (close to or above 300 mm) to cause floods on its own (see Tables 3 and 4).

Similar results were obtained in a study of rainfall patterns and critical values associated with landslides in Povoação County, São Miguel Island, Azores. The findings from the Azores highlight critical rainfall thresholds for landslide occurrence and emphasize the importance of both short-duration, high-intensity rainfall events and long-lasting, lower-intensity rainfall episodes in triggering landslides (Marques et al., 2008).

There is no doubt that the combination of these two variables (antecedent precipitation and precipitation during the event) is indispensable for predicting catastrophic floods. However, even when dealing with relatively high 24-hour rainfall maximums (for instance, exceeding 300 mm), the potential occurrence of floods depends mainly on the hourly variation of that maximum daily quantity. The higher the hourly concentration, the greater the likelihood of torrential flooding. Heavy

rainfall occurring over several hours can indeed trigger catastrophic floods in mountain streams. Episodes lasting less than 24 hours are usually associated with variable amounts of 200 mm to 400 mm or more. Normally, more than 80% of this rain is concentrated in less than 12 hours. On average, about 45% of this precipitation—corresponding to an average accumulation of 150 mm—occurs concentrated in less than 6 hours.

Therefore, the results are sufficiently consistent to propose criteria for predefined rainfall thresholds (Table 5). The thresholds obtained can be enhanced by incorporating rainfall measurement data from future heavy rainfall events.

To summarize, knowledge of calibrated antecedent rainfall is particularly important for identifying critical soil moisture conditions that are likely to generate greater slope instability and consequent sediment inflow into streamlines, leading to an increase in the magnitude of torrential floods. On the other hand, understanding the daily and sub-daily variation of heavy rainfall events can help predict a catchment's response time to peak precipitation.

We should not overlook the limitations of this methodology, which relies on point measurements of rainfall. Therefore, it is crucial to acknowledge the uncertainties and limitations of the results obtained. Rainfall often occurs in a dispersed manner, making it challenging to obtain a sample that accurately represents the geophysical conditions. To obtain more accurate results on the relationship between calibrated antecedent precipitation, maximum daily and sub-daily rainfall and the corresponding runoff peaks, it would be ideal to have at least one rain gauge for each catchment. However, this is not always feasible to implement.

According to the HEC-HMS model results, under streamflow conditions generated on January 7, 2021, the corresponding peak flows at the mouth of the Faial (50 km²) and São Jorge (32 km²) catchments reached 297 m³/s and 215.6 m³/s, respectively. The hydrographs of the terminal sections of the catchments provided evidence of the complex nature of these floods, characterized by several flood peaks synchronized with secondary peaks of precipitation occurring in the preceding two or three hours. Generally, the occurrence of secondary peaks of intense precipitation prolongs flood discharges for extended periods (hours), thereby increasing the risk of catastrophic floods.

However, the peak flood flow estimations obtained from the Hydrologic Engineering Center-Hydrologic Modeling System (HEC-HMS) model may differ from the values obtained with direct measurements in cross sections using channel geometry methods. This discrepancy arises because the model is specifically used for calculating fluvial liquid discharge. But, in reality, torrential floods consist of a mixed flow of solid material and water. Unfortunately, the current torrential flood risk assessment system lacks an important factor: sediment supply (Lane et al., 2008; Wang et al., 2019). While constant sediment inflow from upstream has little effect on the water level during stream flow discharge, sudden sediment inflows from valley slopes, such as debris flows, significantly increase flood risk in a given stream reach, especially near stream junctions (Liu et al., 2022). Sediment transport can indeed lead to higher flood discharge. In reality, there is a complex relationship between sediment transport and flood discharge, with both processes influencing and being influenced by each other. However, for the purposes of comparative analysis between catchments, we considered that the values obtained reflect an order of magnitude sufficient to perceive the spatial impact of floods. Nevertheless, comparing the values estimated by modeling with those obtained by applying the Manning-Strickler Formula, based on direct measurements in cross sections

using channel geometry methods (only available in one of the catchments studied), reveals that the former are underestimated compared to the latter. This flood factor is not the central theme of this research and should be the subject of in-depth investigation in future work.

On the island of Madeira, intense precipitation events, combined with previously identified critical thresholds, will trigger the rapid mobilization of substantial amounts of solid material due to slope movements. This material enters the streamlines, leading to severe torrential floods. Typically, debris flows result in significant alterations to the morphology of streambeds and banks. Additionally, these flows can sometimes obstruct hydraulic passages, which are often undersized to accommodate the passage of solid material. Therefore, in some streams, the flood situation (overbank flow) only occurs due to the presence of artificial obstacles that restrict the passage of high flows. Without these obstacles, flooding would probably not occur in some streams. This is yet another aspect that demonstrates the complexity of the combination of factors that influence floods.

**Conclusions**

According to historical records, heavy rain events can occur anywhere on the island, although they are more frequent in the mountains. Therefore, data from a mosaic of site-specific rain gauges across the island was used. The goal was to extract a combined regional threshold that depicts the behavior of heavy rainfall events, which are responsible for triggering torrential floods on a relatively small mountainous island (741 km²) with short distances between the mountain tops and the coast (less than 15 km).

The research has identified predefined rainfall thresholds for torrential floods based on precipitation from preceding days and weeks, maximum rainfall in 24 hours, and sub-daily rainfall. It was possible to conclude that, in most cases, antecedent rainfall significantly influences the occurrence of torrential floods on Madeira Island, thus validating the selected method as appropriate, given the hydrogeological characteristics of the region. However, in a few cases, the rainfall from the event can be high enough to cause torrential floods, regardless of the preceding rainfall.

On a sub-daily scale, it is interesting to observe a strong correlation between maximum precipitation over 24 hours and that over 12 hours. Consequently, with access to 24-hour numerical precipitation forecast data, it becomes feasible to estimate the maximum precipitation concentrated in less than 12 hours, which has the potential to trigger torrential floods.

In fact, this methodology can be applied in other locations if historical data on heavy rainfall episodes is available. However, the influence of prior rainfall on triggering floods depends primarily on the prevailing types of soils and rocks in the area, as well as their retention capacity in the superficial soil layer. Rainfall from preceding days and weeks can create favorable conditions that increase slope instability and accelerate surface runoff. Understanding the rhythms of rain and its impact on flooding is crucial for disaster preparedness and risk management.

Further refinement of the results is possible with additional data, so maintaining an inventory of heavy rainfall-torrential floods, by incorporating data from future occurrences, is essential to keep the critical thresholds for the system up to date. In our opinion, this study has significantly reduced uncertainty regarding precipitation thresholds in Madeira Island. In this

way, we are contributing to the common objective of any early warning system: the timely detection of heavy rainfall events that exceed specific thresholds. This allows for issuing warnings to the population and preparing a response to minimize the negative effects of torrential floods.

The HEC-HMS (Hydrologic Engineering Center - Hydrologic Modeling System) software was used to estimate flow hydrographs and peak discharges, but without supplemental analysis tools addressing erosion and sediment transport. In this sense, the results obtained have some limitations. In reality, torrential flows are a mixture of water and debris. Further investigation should be done to better understand the temporal and spatial patterns of rainfall-surface and stream runoff events in these catchments. Since the results related to peak discharge are still incipient, future research should focus on the systematization of information resulting from real-time monitoring of fluvial dynamics during flood events.

**Author contribution**

Sérgio Lopes and Marcelo Fragoso conceived the idea for this work. Marcelo Fragoso developed the theoretical framework, while Sérgio Lopes carried out the computations. Sérgio Lopes wrote the manuscript, with support from Marcelo Fragoso and Eusébio Reis. Eusébio Reis supervised the project. All three authors contributed to the final version of the manuscript.

**Acknowledgements.** The authors express their gratitude to the Portuguese Instituto de Meteorologia and the Regional Civil Engineering Laboratory of the Autonomous Region of Madeira, Portugal, for providing data from their meteorological monitoring rain-gauge stations.

This research was funded in whole or in part by the Fundação para a Ciência e a Tecnologia, I.P. (FCT, https://ror.org/00snfqn5816) under Grant (Funder Grant number and/or Grant DOI). For the purpose of Open Access, the author has applied a CC-BY public copyright license to any Author's Accepted Manuscript (AAM) version arising from this submission.

**Declaration of competing interest**

The authors declare that they have no conflict of interest.

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
