# Peer review of "Rainfall analysis in mountain streams affected by torrential floods on Madeira Island, Portugal"

_EGUsphere, 2024_

## Author Response (AR1)

**Point-by-point response to the reviews**

**RC1: 'Comment on egusphere-2024-513', Anonymous Referee #1, 03 Apr 2024**

Dear Author

I reviewed your paper "Local floods in Madeira Island between 2009 and 2021. Rainfall analysis and risk assessment in mountain streams". Despite I recognize that you and your co-authors worked a lot on it, and your English is perfectly understandable, I think that your paper is still not ready for an international journal as NHESS. In the following, I will describe the points that not convinced me and I give you some suggestions to strength them.

First of all, the paper is shaped as a case study, a report of events occurred in a specific site, without any possible relation with other case studies/sites. I suggest to ask yourself what a reader (that is not interested to your study area) could learn from your paper. The methodology? In this case you should strength the methodology description, introducing a flow chart and keeping in mind people that are interested in reproduce the same approach. You should give less emphasis to the description of the effects of the events, which are local, and repeat themselves everywhere, even with stronger severity.

Answer: The flowchart has been drawn up and correspond to Fig. 1 in the new version of the manuscript. I believe that in the new version of the methodology we can find more detailed information on the steps that are followed to obtain critical rainfall threshold values, and, in order to better understand how the temporal variation of rainfall can influence the occurrence of torrential flood events.

It is difficult to appreciate the actual significance of this research because methodology and results are merged and it is difficult to understand what are the premises and what are the results. More in general, you should work to highlight:

**what is the aim of the paper**?

Answer: The primary objective of this study is to determine critical precipitation threshold values that can lead to torrential flows.

**What you are doing in this paper**?

Answer: To achieve this goal, we will base our analysis on historical torrential flow events and past rainfall measurements. Specifically, we will compare data from rainfall events that triggered torrential events with data from those that did not.

**And why**?

Answer: It is worth noting that this study was only feasible due to the availability of high-quality base data. The data includes excellent temporal resolution readings at 10-minute intervals from a network of approximately 40 rain gauges distributed across different locations. These devices provide comprehensive spatial coverage for rainfall monitoring in the catchments of Madeira Island.

**What is the need that this paper is going to meet?**

Answer: This article aims to address an existing gap related to critical thresholds for heavy rain and subsequent torrential flows. In our opinion, this study will significantly reduce uncertainty concerning precipitation thresholds on Madeira Island.

**In what way this will happen?**

Answer: This new data can be used for establishing accurate rainfall thresholds for torrential flow early warning systems.

Answering to these questions could help you to rearrange the paper in a way more similar to the classical structure of scientific papers.

Particularly, Introduction must state the objectives of the work and provide an **adequate background, placing the study in a broad context and highlight why it is important**. It should define the purpose of the work and its significance and the current state of the research field reviewed using key (and recent) publications, also

highlighting controversial and diverging hypotheses. Oppositely, your Introduction is an introduction to the study area and no recent papers with a similar approach are quoted and compared in terms of objectives and methodology to your paper

Answer: The introduction to the article has been revised in line with your recommendations. The main objective of the article is to apply a specific methodology to derive precise information regarding critical rainfall thresholds that may lead to torrential floods. In practice, this involves identifying the critical combination between the rainfall during the event and the accumulated rainfall from preceding days and weeks. This information serves as a crucial input variable for a flood early warning system.

**Methodology** must provide sufficient details to allow the work to be reproduced by an independent researcher

Answer: The methodology was also reviewed, and a flowchart was created to assist any researcher in reproducing it.

**Conclusions** must report the conclusions (not the synthesis) obtained from your study and possible applications in other frameworks, highlighting whether some of them have generic application, rather than being case-specific.

Answer: This topic was also revised accordingly.

About figures: you should do a selection of most significant figures for readers customized to see geomorphic effects of floods and landslides. Captions of figures and tables are very long: I suggest to review them.

Answer: I appreciate the suggestion to select the most significant figures, because in fact we can better visualise the effects of floods through one or two examples.

Tables should be formatted in a better way, avoiding redundancies and rows with more than a line (you should simply enlarge the columns…). And moreover, in an international journal you should try to summarize instead of present long lists of data. Actually, you should identify and propose only data functional to highlight something (i.e. review tab 4).

Answer: The text of the figure and table legends has been simplified as much as possible, and the formatting of the tables has also been revised. In the context of modifying the article's structure, it was decided to delete Table 4.

References denotes the absence of a specific research on recent similar papers. I suggest to do this step before rearrange the paper and to add to references only the articles quoted in the paper (Blöschl is only in references)

I hope that these suggestions can help you to improve your paper.

Answer: The work of researching and updating the bibliographical references was carried out as part of this revision of the manuscript.

**RC2: 'Comment on egusphere-2024-513', Anonymous Referee #2, 05 Apr 2024**

This study is a comparative analysis of torrential floods that occurred in specific villages on the northern side of Madeira Island during December 2020 and January 2021, compared with earlier local events from 2009, 2012, and 2013. The study focuses on understanding the hydrogeomorphological events characterized by interactions between slope movements (landslides, debris flows) and fluvial dynamics (flash floods) in mountainous regions like Madeira Island, triggered by intense precipitation episodes exceeding critical thresholds, leading to peak discharges in various catchments.

Although the study has some interesting insights, it appears to be in need of a lot of improvements and restructuring to make it of a quality that would be ready to be considered for NHESS. **For example, the description of susceptibility appears to oversimplify the relationships between topography, geology, and flood hazard susceptibility.** English writing should be improved on topics like sentence

constructions, and the flow/narrative of the story should be improved a lot for some sections like introduction and discussion.

**Specific comments:**

Suggestion to write title without full stop to make it more attractive.

**Answer:** The new title is as follows: Rainfall analysis in mountain streams affected by torrential floods on Madeira Island, Portugal

**Abstract**

To increase interpretability and readership of the manuscript please keep to the conventional way of writing an abstract (background, rationale, main message, approach/results, implications). For me now the abstract does not give a proper overview of the manuscript.

**Answer:**

background, rationale, main message,

Torrential flows are powerful and destructive events that result from a mix of debris and water moving rapidly down steep channels in mountainous areas. They are normally triggered by heavy rainfall. This combination creates a high-density mass that is displaced by gravitational force, often through successive impulses. Predicting torrential flows is crucial because this natural hazard can pose a significant threat to both humans and infrastructure. Critical rainfall analyses is a fundamental research task with practical applications in early warning systems.

approach

Empirical rainfall thresholds are calculated based on historical local-scale torrential flow events and past rainfall measurements that occurred on Madeira Island between 2009 and 2021. This analysis differentiates between rainfall from previous days and weeks and event rainfall. The first parameter uses a calibrated antecedent precipitation technique based on a power law that accounts for the drainage process over time in the catchments. The second corresponds to the maximum 24-hour precipitation, calculated as a sliding sum over the day(s) of heavy rainfall. The results

from the combinations between these two parameters are considered to establish empirical rainfall thresholds related to torrential flows.

results

Our results statistically demonstrate that calibrated antecedent precipitation was significantly different between triggering and non-triggering rainfalls. Therefore, antecedent rainfall influences the occurrence of torrential floods on Madeira Island. By applying this methodology, we were able to determine predefined rainfall thresholds for the occurrence of torrential floods. Calibrated antecedent precipitation over 15 days (110 mm) and 30 days (130 mm), combined with a maximum rainfall of 250 mm in 24 hours, exemplify critical conditions.

implications

As far as we know, this is the first study to introduce predefined rainfall thresholds for the occurrence of torrential floods in Madeira Island catchments. In this context, the rainfall thresholds, particularly the results we have obtained, serve as a crucial input parameter for establishing a robust torrential flood warning system.

Some sentence are very long and hard to dissect.

Refrain from using words like "surprising".

"Early warning system" is mentioned in the abstract, but not in the manuscript. Why is this mentioned only here? And if so, there should be a clear indication why it is included

**Answer:** The absence of a reference to the early warning system in the manuscript was an oversight, which has now been duly addressed in the new version of the article. The main purpose of critical rainfall calculations stems precisely from the need to have more reliable critical value data for a warning system.

**Introduction**

Header numbering missing

**Answer:** Corrected.

Layout of text is different than rest of the manuscript

**Answer:** Corrected.

What is meant with "Bibliography" line 38?

**Answer:** this word has been removed in the new version of the manuscript.

Impressive has a positive connotation, so maybe use another word when talking about disasters.

**Answer:** these reminders have been taken into account in the new version of the article.

Instead of reading an introduction which builds on top a problem statement with background information and scientific relevancy of the identified research gap, this introduction reads as a case study area description. Suggestion to rewrite introduction. The overall objective should be clearly defined based on the research gap. Now it just states it is expected to bring new scientific contributions. The research objectives are not well identified.

What is meant with analyse the data?

Answer: This analysis is based on rainfall data, particularly 10-minute readings from the rain-gauge network.

What type of analysis?

Answer: The identification of critical thresholds was accomplished through a statistical analysis of precipitation data from years with and without torrential flood records. This analysis aimed to detect a clear separation in precipitation behavior between both sample groups.

How is the relevancy demonstrated of the high-resolution susceptibility map?

What is meant with high-resolution?

Is local-scale or small-scale more suitable?

Answer: In the initial version of the article, the cartography served the purpose of providing a schematic representation of areas affected by known torrential floods. The goal was to present the physical boundaries of recent floods in specific localities, including the regions where debris flows occurred. This was achieved through figures

based on a high spatial resolution base map, potentially supplemented by photographic records. However, during the ongoing article revision process, it was decided to exclude this topic from the present study. The reason behind this exclusion lies in the lack of a concrete methodology for assessing susceptibility to torrential floods, including relevant conditioning and triggering factors. And also because the main objective of this study is to determine critical thresholds for heavy rainfall.

**Data**

To help the reader understand the manuscript better, a flowchart depicting the methodological steps is needed.

Answer: The flowchart has been drawn up and correspond to Fig. 1 in the new version of the article.

The technique is commonly referred to as Annual Maxima

As correctly stated in line 83-83, for impact analysis annual maxima might not be relevant as more than one extreme value per year can be of importance. Why is not a peak over threshold method selected here?

Answer: We chose not to use this method in our analysis because we believe it is more realistic to base our methodology on a sample of rainfall data related to known real cases of heavy rainfall events that have occurred in the mountainous regions of the island and have led to torrential floods.

Please use subheadings for sections throughout the manuscript to help the reader understand the aim of the specific section.

Answer: This suggestion was taken into account during the revision of the text.

**Line 91, what is meant by maximum precipitation variable in 24 hours? What is the unit of this variable?**

Answer: The maximum rainfall in 24 hours corresponds to the highest amount of rainfall that occurs within the 24-hour moving interval, calculated based on the 10-

minute readings from the udographs on the day(s) of the rain event. The unit of measurement is always in millimetres (mm).

**What would be the sensitivity of using k=0.8 to the findings of the study?**

Answer: In theory, with the empirical parameter K = 0.9, we assume that 90% of the rainwater that occurred in a certain period prior to the day of interest (even before applying the exponent) remains accumulated in the soils of the catchment-stream system. This evaluation considers rainwater from previous days and weeks, which can contribute to soil saturation, instability, and soil movement. The remaining 10% of that water has already been drained into the water network, percolated, or evaporated.

If the option had been K = 0.8, it would have meant assuming that on steeper slopes (such as narrow-bottomed valleys, as found on Madeira Island), a higher percentage of rainwater (in this case, 20%) quickly runs off the slopes into watercourses or is lost through evaporation or percolation. Since this is an empirical parameter, there is naturally some subjectivity in choosing its right value.

**Rainfall**

Please also show R2 values for Figure 6

Where there checks done for homoscedasticity of the data?

The plots are difficult to interpret and compare when the x-axis scale changes each subplot in Figure 6

Answer: After submitting the article, all the calculations were revised and an error was detected in the application of the mathematical formula for calibrated antecedent precipitation calculations. Consequently, all the calculations be had to be remade. This process has now been completed, and the arrangement of data pairs in the graphs has been reorganized. With this new arrangement, it no longer makes sense to apply the rules for the equations of the regression lines. Instead, for the different durations shown in Figure 6, new thresholds for maximum precipitation in 24 hours and calibrated antecedent precipitation were identified, using only absolute specific and predefined values of the two variables. These updated values are presented in the corrected version of the article.

The x-axis scale has been standardised.

The rainfall data samples used in this analysis correspond to relatively short climatic series. Therefore, it was decided not to apply the homogeneity tests, normally used for long series of at least 30 years.

**Discharge**

How does the occurrence of secondary peaks of intense rainfall correlate with the complex flood patterns observed in the studied catchments?

Answer: As a general rule, the occurrence of secondary peaks of intense precipitation contributes to prolonging flood discharges for longer periods (hours), and thus increase the risk of catastrophic floods. In other words, these floods are complex because they result from several intense downpours, which tend to increase the instability of the soil in the upper sectors of the catchments.

How reliable and valid are the peak flood flow estimations obtained from the HEC-HMS model?

Answer: The HEC-HMS (Hydrologic Engineering Center - Hydrologic Modeling System) software was used to estimate flow hydrographs and peak discharges, but without supplemental analysis tools addressing erosion and sediment transport. In this sense, the results obtained have some limitations. In reality, torrential flows are a mixture of water and debris. However, for the purposes of comparative analysis between catchments, we considered that the values obtained reflect an order of magnitude sufficient to perceive the spatial variations of floods. Nevertheless, comparing the values estimated by modeling with those obtained by applying the Manning-Strickler Formula, based on direct measurements in cross sections using channel geometry methods (only available in one of the catchments studied), reveals that the former are underestimated compared to the latter.

**Flood risk**

Line 436: Susceptibility to?

I find the susceptibility map to be oversimplified as it also seems to be just an indication of previous events. There is limited quantitative analysis or data-driven assessment of susceptibility factors such as slope gradient, geology, rainfall intensity, or land cover. Without quantitative analysis, the susceptibility map may not accurately represent the true spatial distribution and intensity of flood hazards.

The text indicates that the susceptibility map was created based on field survey work and manual identification of areas susceptible to torrential floods, slope movements, and debris flows. The term "cartographic sketch" suggests a less formal and more subjective approach to mapping susceptibility.

Answer: In fact, that was the whole point, to create a less formal and more subjective approach to mapping susceptibility. Let's say that this more simplified approach was intended to provide land-use planners with a source of spatial information on hydrological hazards, based essentially on the historic of occurrences in a given part of the territory, thus making a clear distinction between potential susceptibility and actual susceptibility. However, as part of the in-depth revision of the initial manuscript, at this stage of the research, it was decided to mainly explore the subject of heavy rainfall thresholds, and to remove the analysis of susceptibility to torrential floods from this article.

Indeed Figure 6 is not precisely a map of susceptibility to torrential flooding. Instead, its purpose was to create a schematic figure illustrating the areas affected by the flood that occurred on December 25, 2020. The intention was to demonstrate that in mountainous regions, even in very small catchments (1 km²), floods can pose a significant risk.

**Discussion**

The discussion appears to be a summation of what was found in the study. There is no discussion on limitations or implications of this study. It would help if this study has a

clearly defined research question to answer, which was also lacking in the introduction.

Answer: The primary objective of this study is to determine critical precipitation threshold values that can lead to torrential flows. To achieve this, we will base our analysis on historical torrential flow events and past rainfall measurements. Specifically, we will compare data from rainfall events that triggered torrential events with data from those that did not. A secondary objective is to obtain complementary hydrological information regarding the relationship between heavy rainfall amounts and flow discharges in small catchments, achieved through a hydrological modelling exercise. The third goal is to discuss aspects related to the uncertainty surrounding the definition of rainfall thresholds and consequent flow discharges. In this context, it is important to consider the uncertainties and limitations of the results obtained, given the limitations of rainfall point measurement methods. Rainfall often occurs in a dispersed manner, which makes it challenging to obtain a sample with meaningful physical representation. Even so, the results obtained are certainly a valuable contribution to improving an early warning system for torrential floods.

**List of all relevant changes made in the manuscript.**

1 Total revision of the abstract;

2 Total revision of introduction;

3 Partial revision of the methodology;

4 Modification of the structure of the presentation of results and discussion

5 Removal of the topic on susceptibility;

6 Total revision of the conclusion;

7 In-depth research and review of bibliographical references.;

---

## Referee Report (RR1)

In the manuscript 'Rainfall analysis in mountain streams affected by torrential floods in Madeira Island, Portugal', the authors conduct a statistical analysis between different precipitation indicators of different time scales to identify critical thresholds relevant for torrential flows. This is potentially an interesting contribution to the academic literature, but requires some significant amount of clarifications and re-organization of the information to make it more comprehensible. I thus propose the following major revisions.

**Abstract:**

- According to the NHESS submission guidelines, an abstract should be "be short, clear, concise", which is commonly translated into 150-250 word limits. The current abstract has more than 400 words, and contains a lot of details not relevant to convene the key message of the paper. I would thus suggest that the authors shorten the abstract to increase the accessibility of this manuscript to the scientific community.

**Introduction:**

- The introduction starts off well explaining the concept of torrential flows and why (antecedent) rainfall events are critical to identify the risk of torrential flows and what the objective for the study is.
- However, lines 67-78 seem totally unrelated, discussing needs for rainfall measurement networks and either need to be better integrated or left out.
- Instead, authors could add a paragraph introducing the case study area and why they specifically look at torrential flows there. Are there numbers that support the criticality of this hazard? Is climate change threatening an increased risk of torrential flow, etc.

**Data and Methods:**

- This section could significantly be improved for readability if the authors would introduce clear subsections (and potentially subsubsections?).
    - One introducing and characterizing the case study area (which is currently done in between at different places in the manuscript),
    - another one introducing the full set of collected data (the current set-up is very confusing at it not clear how and why different sets of rain gauges or specific events were used for the results in section 3.1 and 3.2 and 4),
    - one on the methods applied for the analysis where additionally to the CAPx equation (I like how the authors introduce the equation in 123, the power is now correctly shown?) methods to determine the 24h-max etc could be clearly described as well as the hydrological model and used equations.

- The clarity of the manuscript/method would benefit if the visualization of the methodology would reflect the elements of the sections. Similarly, to be a flow, it would be good to provide some sense for direction and ordering.

**Rainfall Thresholds (Results):** This chapter offers a lot of fruit for thought and some minor comments (see below).

- One major unclarity is how Figure 4 relates to the used input data. The authors mention that there are 7 torrential flow events in the considered time horizon (without 2023) and compare these values against annual Pmax24h of years without (so 5 years left?). It is not clear how these events can be found back in Figure 4. There seem to be 13 orange dots, 11 light blue and 12 dark blue. Clarification what they mean and how they are related, would help support the claims made by the authors. Similarly, it is not clear what the vertical line represents.
- Regarding section 3.2: It is unclear to me, what the authors intend to do with this correlation analysis between different temporal scales in the context of torrential flow. While I was expecting that the authors might explore whether the choice of Pmax24 is accurate to predict torrential flows, the authors seem to go the other way around and discuss whether Pmax24 is well correlated with other PmaxT. Readers need more guidance in this section (and prior) to understand why this analysis is done and why it is relevant.

**Estimation of peak discharges:** It is unclear why this analysis is conducted. Furthermore, the authors could consider visualizing the data of max discharge dependent on catchment size and/or Pmax24. The current way of presenting the results in descriptive text makes it very hard to discover any sort of patterns. Again, clarfiying why specific events are chosen for the analysis (and not all) would be an important information to add.

**Discussion:** I would encourage the authors to reorganize the discussion section to make clear what current limitations are of the study and how the findings of this study relate to findings of the research community. At times, it reads as part of the introduction of the case study (e.g. l. 451-461, 462-470, l.480-486). While other statements could benefit from more elaboration/linking to the results (e.g. l. 476-477, l.530-533) or unrelated to the topic of the manuscript (e.g. l.487-489, l.508-509). It would also be interesting to have a reflection on Table 5 which suggests that no matter what PAC is used, the Pmax24 threshold is always the same. Similarly, the discussion of uncertainty and limits of the study would be valuable to reflect on remaining research gaps.

**List of minor comments:**

- L.81: How is the secondary and tertiary objective linked to the information about torrential flows that has been the focus of the introduction?)
- L.123: The factors for P should be as a power, right?
- L.126: What is the sensitivity of choosing k=0.9 compared to k=0.8? Would it have influence on the proposed thresholds? The authors write later about low permeability of the study area (l.154), how does this relate to the choice of a high k value? Doesn't low permeability mean that water run-off is much higher, so less water infiltrates?
- L.214: Why was it necessary to include the rainfall event from 2023?
- L. 216: The auhors mention that a statistical analysis is necessary for precipitation data froam years with and without torrential flood records to detect patterns. It seems however, that the authors have done the analysis mostly by means of visual analysis using Fig. 4. More elaboration would be helpful to clarify how the statistical analysis between different years has been done.
- L.210: Why are the two events from 2011 not in Table 2 as well? Should the 2023 event also be added there (no information whether torrential flow occurred)
- **L.243:** Using the case of Dec/20 and Jan/21 is a bit misleading. The authors state in regards to the Dec/20 'daily and sub-daily maximum precipitation values (Table 3) were sufficiently high to cause catastrophic floods'. At the same time, the 2021 torrential event has even higher Pmax24 so following that logic, we cannot say that CAP15 plays a role here or not. Would be good to clarify.
- L. 295: Fig5 the axis are not readable.
- L.312: Why did the authors disregarded the events in 2010 and 2011 (and 2023) for the analysis?
- L. 320: I strongly advise the authors to have a clear input data section in their introduction to explain consistently which data-sets are used for what and why those data-sets were chosen.
- L.345: Is figure 6 now using the information from torrential flow events, all inputs from between 2009 and 2021 or the extended data-set with the extra 6?
- L.365: Why are the authors only reporting these information for the five events and not the full set of 11 (?)
- L.550: I don't understand the asterisk. Has this been discussed beforehand? How do the authors make this claim?

---

## Author Response (AR2)

**Point-by-point response to the reviews: October 2024**

Report #1

Submitted on 01 Jul 2024

Anonymous referee #2

**Suggestions for revision or reasons for rejection**

(visible to the public if the article is accepted and published)

Dear authors,

I appreciate all the work done and see that the manuscript has been improved a lot. All my comments are addressed, yet a few of them remain up for discussion. Therefore, I recommend that the following issues should be addressed upon consideration for publication.

- The addition of a flowchart is much appreciated, however the output of the manuscript is not a EWS. The authors rightfully argue that the output can be used for EWS, but now the flowchart is misleading. The EWS part can be for the discussion but is not part of the metholdology and not an end product.

Answer

Thank you for your comment, because this is indeed an extremely assertive reminder. In fact, the methodology adopted results in a combination of threshold values being exceeded. The graph has been updated accordingly.

- I see indeed that the annual maxima has been removed from the manuscript. The authors now argue to only include real cases of heavy rainfall. The same question holds here as well. How has this been selected? What criteria has been used? And why?

Answer

In years without records of torrential floods, the precipitation values needed to analyse the relationship between daily precipitation and antecedent precipitation were obtained using the maximum annual precipitation technique for various durations.

However, when constructing these extreme value series for years with records of torrential floods, multiple values per year can be utilized. This becomes particularly

relevant when two heavy rainfall events followed by floods occur within the same hydrological year.

The identification of critical thresholds requires a statistical analysis of precipitation data from years with and without torrential flood records, to detect the existence of a clear separation in precipitation behaviour between both samples.

- To understand correctly, the authors selected 24 hour max rainfall to be the highest value of the hours after having applied the 24 hour moving statistics. Is this a moving average? If so, this measure highest rainfall amount in 24 hours time window if taking the max. However, response time of catchments appear to be 2 hours, which makes for an interesting discussion as it brings the question, what determines added values of 24h peak. How important is the contribution of a wetter system and how system memory can contribute to the results. Such a discussion or analysis would be an interesting addition. In addition, Table 3 is difficult to compare and I would suggest standardizing the values someway for easier comparison.

Answer

In fact, it is always the sliding sum of the 10-minute rainfall records over a 24-hour period. After applying this calculation, the maximum in 24 hours is automatically obtained.

To understand the response time of catchments, we needed to look at the sub-daily variation in rainfall and in particular the hourly peaks. Figure 8 shows the results of this comparative analysis (hyetographs).

- Lastly I noticed that not all the clarifications, the authors nicely did in their author response, was added in the manuscript. Those clarification might also guide other readers that have the same questions (for example K values explanation, discharge patterns, and reliability HEC-HMS)

Answer

In fact, it makes perfect sense to insert all the clarifications regarding K values explanation, discharge patterns, and HEC-HMS reliability. It was an inadvertent failure

on our part not to insert them. The new version of the manuscript already includes these clarifications.

Report #2

Submitted on 19 Sep 2024

Anonymous referee #3

**Suggestions for revision or reasons for rejection**

(visible to the public if the article is accepted and published)

I have been invited to review the revised version of the manuscript. Here are my suggestions. The authors have prepared a point-by-point response to the reviewer's comments but did not mention where they have made the changes in the revised manuscript. It is essential to note the line number or at least the page number below each response so that reviewers can easily follow the amendments in the revised manuscript.

In addition, the revised manuscript and the track changes version both look identical to me. I cannot trace the changes made to the original manuscript in response to the comments by reviewers. I request the authors to prepare a track changes version of the manuscript highlighting the changes they made. I would be happy to review the revised version again. Thanks!

Answer

With regard to this issue, the main changes made compared to the original version have been signalled in the new version of the manuscript, and all the changes resulting from the new comments from the two reviewers have also been signalled.

However, we ask for your understanding on this matter. It's difficult to identify all the changes made to the original document because the second version of the manuscript has undergone a profound change in its structure.

---

## Author Response (AR3)

**Referee #4**

Your comments were very assertive and precise, so we thank you for your commitment to a thorough revision of the article.

The major revisions indicated by the reviewer for the different topics are all included in the new version of the manuscript. Responses to minor comments are as follows.

**List of minor comments:**

L.81: How is the secondary and tertiary objective linked to the information about torrential flows that has been the focus of the introduction?)

The objectives have been revised.

L.123: The factors for P should be as a power, right?

The mathematical formula was corrected. It was a formatting error.

L.126: What is the sensitivity of choosing k=0.9 compared to k=0.8? Would it have influence on the proposed thresholds? The authors write later about low permeability of the study area (l.154), how does this relate to the choice of a high k value? Doesn't low permeability mean that water run-off is much higher, so less water infiltrates?

It should be noted that K is a sensitivity parameter. Naturally, opting for K=8 would be sufficient to influence the proposed thresholds.

The question of the relationship between the K parameter and the permeability of the land (rock surfaces and soils), in the context of determining the weight of previous precipitation (days and weeks) in triggering floods, is complex.

Indeed, low permeability mean that water run-off is much higher (especially during heavy rainfall events), but the surface layers of the land are mostly made up of soils with an average thickness of up to 1 metre. In soils with these characteristics, rainfall over several days and weeks (less intense, but abundant) tends to contribute to soil saturation.

The calibrated antecedent rainfall equation was created to determine whether it rained a lot or a little in the days and weeks prior to a given moment and what influence this rainfall has on the saturation level of the soil. The K parameter within it results from the combination of various factors such as the lithology and permeability of the terrain, the inclination of the slopes, and vegetation cover. In fact, both the K parameter and the CN curve number parameter aim to recreate the specific soil saturation conditions at a given time.

L.214: Why was it necessary to include the rainfall event from 2023?

In fact, the June 2023 event did not give rise to significant torrential flooding, nor does it influence the data pattern in the graphs in figure 4, so it will be removed in the new version of the manuscript.

L. 216: The auhors mention that a statistical analysis is necessary for precipitation data froam years with and without torrential flood records to detect patterns. It seems however, that the authors have done the analysis mostly by means of visual analysis using Fig. 4. More elaboration would be helpful to clarify how the statistical analysis between different years has been done.

The following information has been added to the methodology: In the case of the sample of years with no record of intense precipitation events—torrential floods—the maximum values of the hydrological year were taken, referring to the maximum 24-hour precipitation (and corresponding CAP value) and the maximum CAP (and corresponding 24-hour precipitation value).

L.210: Why are the two events from 2011 not in Table 2 as well? Should the 2023 event also be added there (no information whether torrential flow occurred).

Information corrected by creating a new table (Table 1).

**L.243:** Using the case of Dec/20 and Jan/21 is a bit misleading. The authors state in regards to the Dec/20 'daily and sub-daily maximum precipitation values (Table 3) were sufficiently high to cause catastrophic floods'. At the same time, the 2021 torrential event has even higher Pmax24 so following that logic, we cannot say that CAP15 plays a role here or not. Would be good to clarify.

The paragraph has been revised accordingly:

The information summarized in Fig. 5 pertains to a specific case involving two events of heavy rainfall-to-runoff torrential floods. These events occurred within the same hydrological year (2020/2021). In this particular scenario, the calibrated antecedent precipitation (CAP) at 5 and 15 days on the date of the first event (December 25, 2020) was relatively low. During the second event (January 7, 2021), the precipitation from the first event significantly influenced the high value of the CAP at 15 days (158 mm). However, the January 7, 2021, torrential event has even higher Pmax24 than the previous event. Thus, in this specific case, it is not possible to ascertain the role of CAP15 in triggering the flood.

L. 295: Fig5 the axis are not readable.

The size of the characters has been increased.

L.312: Why did the authors disregarded the events in 2010 and 2011 (and 2023) for the analysis?

L. 320: I strongly advise the authors to have a clear input data section in their introduction to explain consistently which data-sets are used for what and why those data-sets were chosen.

A new table was created to fulfil this purpose at the beginning of the manuscript. The new information contained in this table clarifies the data samples used.

L.345: Is figure 6 now using the information from torrential flow events, all inputs from between 2009 and 2021 or the extended data-set with the extra 6?

Figure 6 is using data from the set of 11 heavy rainfall events listed in the new Table 1. In this new table, a column has been created to indicate the purpose of the data being

considered. For some heavy rainfall events, data from more than one rain-gauge is taken into account.

L.365: Why are the authors only reporting these information for the five events and not the full set of 11 (?)

There are no records of maximum sub-daily rainfall for all the heavy rain events, but it was possible to add data from the event on 20 February 2010 to the table 4.

L.550: I don't understand the asterisk. Has this been discussed beforehand? How do the authors make this claim?

This issue has been corrected in the new version of the manuscript (2 Data and methods/ The case study area), in order to relate the information from the data samples (Table 1) to the critical thresholds shown in Table 5.

---

## Author Response (AR4)

**Submitted on 29 Jan 2025**

**Anonymous referee #4**

Once again, we would like to thank you for your commitment to reading the manuscript carefully.

Suggestions for revision or reasons for rejection
(visible to the public if the article is accepted and published)
The authors did a good job in addressing previous feedback.

I still think that the paper and its message would benefit from a stronger red thread through the methods and result sections, which still seem a bit like a compilation of separate aspects. I would thus suggest that the authors take a moment to introduce the methodological flow based on Figure 1 (instead of only referring to that Figure) and clarify the reference to this flow and main purpose of that process step throughout the result sections.
Changes made accordingly.

Similarly, while the authors significantly improved the clarity about which data points are used to derive which results, the use of the specific event series (Dec 2020 - Jan 2021) still seems to disrupt the flow as additional and new information are introduced. It might be beneficial to introduce the event series as a sub-section in the section outlining the case study (along with the relevant figures or descriptive data), so that the authors can focus on the key results that were confirmed/derived from that case in the result section.
The manuscript has been amended in line with this suggestion.

Furthermore, there are some minor comments:

Lines 138-139: The meaning of K=8 is unclear. So far, the authors have only discussed K=0.8–0.9 - is this a typo?

This paragraph has been revised to clarify the meaning of the K parameter.

Figure 1: The figure appears incomplete, with some lines missing. Please ensure all elements are properly displayed.

The figure has been revised.

Figure 5: The x-axis extends beyond the key window of interest (Dec 2020 – Jan 2021). Adjusting it to this timeframe could improve readability and may even render Figure 7 redundant.

The reason for widening the time window of analysis was to better understand the temporal variation of daily precipitation and accumulated antecedent precipitation on days with heavy precipitation events and on days without heavy precipitation events. The two figures refer to data from different time intervals. Figure 5 refers to the precipitation of days and weeks and figure 7 to the variation in hourly precipitation.

Line 280 (new text added): The placement of this paragraph is unclear. It seems more appropriate in Section 4 (Relationships between sub-daily precipitation peaks and peak discharges), as it discusses a specific event (or series). Otherwise the motivation for the paragraph should be made more clear.

Corrected accordingly.

Positioning of Figures and Tables: The placement of figures and tables could be improved for better readability. Currently, references to figures are made before the reader has a chance to view them, sometimes introducing multiple figures in consecutive paragraphs without a clear order. Moving figures and tables closer to their first mention would enhance clarity and flow.

The positioning of figures and tables has been changed to ensure greater proximity between the reference in the text and the respective element.